# Multi-Scenario Species Distribution Modeling

**DOI:** 10.3390/insects10030065

**Published:** 2019-03-01

**Authors:** Senait D. Senay, Susan P. Worner

**Affiliations:** 1GEMS™—A CFANS & MSI initiative, University of Minnesota, 305 Cargill Building, 1500 Gortner Avenue, Saint Paul, MN 55108, USA; 2Department of Plant Pathology, University of Minnesota, 495 Borlaug Hall, 1991 Upper Buford Circle, Saint Paul, MN 55108, USA; 3Bio-Protection Research Centre, Lincoln University, 7674 Lincoln, New Zealand; sue.worner@lincoln.ac.nz

**Keywords:** invasive insect species, model uncertainty, multi-model framework, non-linear principal component analysis, principal component analysis, random forest, species distribution models

## Abstract

Correlative species distribution models (SDMs) are increasingly being used to predict suitable insect habitats. There is also much criticism of prediction discrepancies among different SDMs for the same species and the lack of effective communication about SDM prediction uncertainty. In this paper, we undertook a factorial study to investigate the effects of various modeling components (species-training-datasets, predictor variables, dimension-reduction methods, and model types) on the accuracy of SDM predictions, with the aim of identifying sources of discrepancy and uncertainty. We found that model type was the major factor causing variation in species-distribution predictions among the various modeling components tested. We also found that different combinations of modeling components could significantly increase or decrease the performance of a model. This result indicated the importance of keeping modeling components constant for comparing a given SDM result. With all modeling components, constant, machine-learning models seem to outperform other model types. We also found that, on average, the Hierarchical Non-Linear Principal Components Analysis dimension-reduction method improved model performance more than other methods tested. We also found that the widely used confusion-matrix-based model-performance indices such as the area under the receiving operating characteristic curve (AUC), sensitivity, and Kappa do not necessarily help select the best model from a set of models if variation in performance is not large. To conclude, model result discrepancies do not necessarily suggest lack of robustness in correlative modeling as they can also occur due to inappropriate selection of modeling components. In addition, more research on model performance evaluation is required for developing robust and sensitive model evaluation methods. Undertaking multi-scenario species-distribution modeling, where possible, is likely to mitigate errors arising from inappropriate modeling components selection, and provide end users with better information on the resulting model prediction uncertainty.

## 1. Introduction

Various species distribution models have been used to predict suitable insect habitats. Many studies are based on correlative models that use species presence along with environmental data to infer suitable habitats for the species under study [1]. Currently, discrepancies between model results represent a major issue in ecological modeling, and the need for quantifying model uncertainty has been repeatedly discussed in the literature [2,3,4,5,6,7,8,9,10,11,12]. A few important studies have investigated sources of uncertainty in SDMs [6,7,9,10,12], and some have developed techniques for quantifying the uncertainty associated with modeling species’ distribution in ecology and the wider spatial modeling context [6,13,14].

Variation in the predictive accuracy of different models is usually attributed to the inherent robustness of modeling algorithms. Simple models, for example, the bioclimatic analysis and prediction system, BIOCLIM [15] and the point-to-point similarity metric system, DOMAIN [16], have been reported to be suitable for predicting the distribution of rare species occupying a limited environmental niche, representing simple linear interactions among environmental variables. Complex models, such as Support Vector Machines (SVMs) [17] and Artificial Neural Networks (ANNs), with complex functions that consider non-linearity and a large number of variables can handle complex interactions within a multidimensional variable space [5,18]. Jiménez-Valverde, et al. [19] and Chefaoui and Lobo [20] argued that the above claim is not entirely true due to inappropriate comparison of model prediction results for species with varying relative occurrence areas in the above studies. However, there is general agreement of an inherent difference in the predictions of simple and complex models based on differences between model predictions in studies that compare models using the same occurrence data and study area [21]. 

The use of geo-environmental variables in addition to purely climatic variables such as temperature and precipitation, has been reported to increase prediction accuracy in SDM models [1,22,23,24]. However, many studies still depend on using a limited number and type of variables, for example, variables derived only from temperature and precipitation data without additional climatic or geo-environmental information, such as elevation, that might help a model better discriminate an ecological niche [25]. Often, the spatial variation of geo-environmental variables is much higher than climatic variables, and can help characterize unique habitats when used along with climatic variables [24]. General reluctance to using additional environmental variables may be associated with data inconsistency that can occur when using data compiled from different sources and multi-scale variables. As the types and number of variables increase, the likelihood of obtaining variables from multiple sources (e.g., sensors) increases. Moreover, the scales of multi-source variables are also likely to differ, thereby significantly increasing modeling effort. Another, significant complication from increasing the number of variables is the increasing complexity of interactions among large sets of variables. Climatic variables frequently used in a number of species’ distribution studies are assumed to have linear relationships. However, linear relationships are not always observed between environmental variables, especially if large sets of predictors from multiple scales and data sources are used in the modeling process. Therefore, the type and number of variables used can further increase the divergence of predictions between simple and complex models because of interactions among environmental variables used [10,19]. Moreover, any dimension reduction performed on predictor datasets may also affect the predictive accuracy of SDMs. Numerous dimension-reduction methods have been used for various applications, but only a few have been successfully adopted in ecology [9,26,27]. When using multi-sourced environmental predictor variables in combination with climatic variables, it is important to consider the possibility of complex and non-linear interactions between variables before choosing a dimension reduction method. For instance, a multidimensional predictor dataset that may not be adequately represented by a simple model may be successfully modeled after application of appropriate dimension-reduction methods [28]. Therefore, in addition to differences in the nature of the species-occurrence data, choice of predictors, and the models used for prediction, data pre-processing methods, such as dimension-reduction techniques or the method used for variable selection, could affect the performance of a SDM prediction.

The purpose of this study was to investigate the effect of modeling components such as predictor datasets, dimension-reduction methods, and model types, and their interactions on SDM model performance using a factorial experimental design and selected case studies. We further investigated whether it is possible to increase model accuracy by using appropriate predictors, dimension-reduction methods, and model types that better explain the spatial distribution of presence points, and their pattern in the predictor feature space. Finally, we discuss the advantage of using a multi-scenario modeling framework to provide information about model prediction uncertainty.

## 2. Materials and Methods

### 2.1. Geographic Extent of Study Area

This study was carried out over a global scale, therefore, all environmental predictors had global coverage. The different species-presence datasets, however, cover varying spatial extents. All datasets have a resolution of 10 arc minutes (0.17°, 18.5 km at the equator). The secondary geographic focus of the study was the extent covered by the North Island, South Island, and Stewart Island of New Zealand.

### 2.2. Predictor Data

Three predictor datasets were used (Table 1), 1) BIOCLIM19 (P1) consisting of 19 temperature- and precipitation-related variables derived from the WORLDCLIM dataset [29,30]; 2) BIOCLIM35 (P2) consisting of the BIOCLIM19 variables and additional 16 radiation and water-balance soil moisture index-related variables accessed from the CLIMOND dataset [31]; and, 3) the BIOCLIM35+T4 (P3). In P3 there was a set of four topographic variables (elevation, slope, aspect, and hill-shade) derived from a digital elevation model (DEM), downloaded from the WORLDCLIM data portal [29,32], which were added to the P2 dataset. The variables, slope, aspect, and hill-shade were calculated from the elevation data using ESRI’s ArcInfo^®^ spatial analyst software. A 3 × 3 pixel focal area was used to process all three DEM-derived topographical variables. Detailed information on the development of variables in the P1 and P2 datasets is given in Hijmans, et al. [32] and Kriticos, et al. [31], respectively.

### 2.3. Dimension Reduction

Three dimension-reduction methods were used: first, a variable selection with the random forest algorithm (RF; DR1). The RF algorithm can handle large numbers of variables and it is widely used in species distribution modeling. The random forest classifier results in low-bias selection by averaging over a large ensemble of high-variance but low-correlation trees [33]. The Akaike information criterion (AIC) was used to rank variable importance; second, principal component analysis (PCA; DR2) was used. PCA is a mathematical method that transforms a set of raw variables into linearly uncorrelated variables by mapping the newly transformed data on artificial orthogonal axes [34]. The PCA itself, or slightly modified versions of it, have been used in ecological modeling either as a dimension-reduction method or as the main species distribution model [35,36]. The third dimension reduction method was non-linear principal component analysis (NLPCA; DR3) in the form of a hierarchical NLPCA (h-NLPCA), a neural network model developed by Scholz and Vigario [37]. The method is reported to be the true non-linear extension of the linear PCA [37]. The h-NLPCA achieves a hierarchical order of principal components similar to the linear PCA (Appendix B) and is both scalable and stable similar to the linear PCA method [38]. Most h-NLPCA parameters were internally computed as they are adjusted throughout the iterative learning. For this study, network weights were initialized at random. The weight decay was set at 0.001 with the maximum iteration conditionally set by either five times the number of observations or 3000, whichever was minimum.

### 2.4. Species Data

The worldwide distribution of five insect species: (i) *Aedes albopictus* (Skuse, 1894), (ii) *Anoplopis gracilipes* (Smith, 1857), (iii) *Diabrotica virgifera virgifera* (LeConte, 1868), (iv) *Thaumetopoea pityocampa* (Denis & Schiffermuller, 1775), and (v) *Vespula vulgaris* (Linnaeus, 1758) (established in New Zealand), was compiled from three sources. The sources included: (a) the Global Biodiversity Information Facility (GBIF) database, (b) previous literature, and (c) personal communication with domain experts (Figure 1). The geographical extents covered by the presence locations for these species vary widely. Variation in the relative occurrence area (ROA) among these species was important to investigate if the distribution range of a species affects the predictive accuracy of SDMs. *A. albopictus* and *V. vulgaris* have a relatively large ROA (Figure 1A,F), whereas, *D. v. virgifera* and *A. gracilipes* cover an intermediate global extent (Figure 1B,C). *T. pityocampa* (Figure 1E) has the smallest occurrence cover, hence, the smallest ROA (See Appendix A for description on the native and invaded ranges of these five insect species). 

Pseudo-absences were generated using a 3-step pseudo-absence generation method [21]. The first step involves choosing a geographical distance to bind or constrain background data from which pseudo-absences were selected. A measure of how far the correlation structure among variables is conserved through the environmental space using presence points as a reference is used to determine the appropriate distance. The distance at which variable importance changes is used as an indicator of correlation structure change. Variable importance ranking is done by carrying out PCA on successive datasets extracted at incremental geographical distances from presence points [21]. At the second step, environmental profiling is used to discriminate environmentally dissimilar portions of the background dataset bound at a distance identified in step one. Lastly, in the third step, a number of pseudo-absences that balance the number of presences, are selected by k-means clustering the class that is most environmentally dissimilar from presences in step two (refer to Senay, et al. [21] for the details on how to generate pseudo absences using the 3-step pseudo-absence selection method. Iturbide, et al. [39] describe a slightly different methodology). The correlation structure among variables changes as the occurrence points-predictor data combinations change making the variable importance over distance different [21]. Therefore, three distances were calculated (Table 2) for each species modeled using the three different predictor datasets (P1, P2, and P3).

Pseudo-absence selection was carried out after the background data was projected onto the Mercator projected coordinate system and interpolated to an equal area grid. The equal area grid projected data was chosen so that any poleward bias that might occur by selecting pseudo-absences within a buffer defined in the unequal grid geographic coordinate system is avoided. The selected pseudo-absence points were re-projected onto a geographic coordinate system for the remainder of the modeling process (Figure 2). 

### 2.5. Model Type

An exhaustive comparison of all available SDMs would be impractical. In this study, we selected four model types to provide predictions for the global distribution of each species based on various P and DR combinations. The first model is quantitative discriminant analysis—QDA (MT1): discriminant analyses in general, and QDA in particular, are classical multivariate models used in species distribution modeling [10,40,41]. While QDA allows easy assessment of variable contributions and assessment of the species distribution prediction, it cannot handle datasets where the number of observations is smaller than the number of variables. The qda function from R [42] and MASS [43] libraries was used to run QDA in the multi-model framework. The second was logistic regression-LOGR (MT2). LOGR is one of the most frequently used SDMs for species distribution studies [6,44]. The glm function in the Stats [42] package in R was used to run LOGR in the multi-model framework. The third model was classification and regression trees—CART (MT3). CART is a classification and regression decision tree that is also frequently used in species distribution models [40,45]. It has been suggested that decision trees incorporate the complexity needed to explain interactions between multidimensional variable data, without a complicated rule that can be easily explained to end-users [40]. The rpart package for R was used to run CART in the multi-model framework. The fourth was a support vector machine—SVM (MT4). Support vector machines are models based on machine learning theory, specifically, artificial neural networks [17]. SVM has been shown to be an excellent classifier in a number of disciplines, for example, in astronomy, medicine, physics, and pattern recognition [46]. SVM has recently been used in ecological modeling along with other machine learning methods such as boosted regression trees (BRT) and artificial neural networks (ANNs) [40,44]. The SVM model has the option of fitting data either linearly or non-linearly. The specific functions used were linear, radial basis, and polynomial. The Kernlab [47] package for R was used to run SVM in the multi-model framework. A separate optimization of the model was carried out to identify the best parameters for the different datasets. The multi-model framework developed by Worner, et al. [41] was used to run the four models in a standardized set-up. Model parameterization and data exporting was also done using this framework.

### 2.6. Research Design and Model Conceptualization

Presence data for five species (PD) were used, and sets of pseudo-absence (PA) data were developed for three predictor datasets (P) and three dimension reduction methods (DR) comprising, 5 PD × 3 P × 3 DR = 45 PA datasets (Figure 3). Finally, 45 training datasets (TD) were prepared by combining PD and PA datasets for each PD-P-DR combination. TD datasets for each species were used to train and evaluate four different models (MT). Within each TD dataset, 80% of the data was used for model training and 20% for model evaluation. Each run was replicated 20 times. Each MT was used to predict the global distribution of each species based on the various P and DR combinations comprising, 3 P × 3 DR × 4 MT = 36 different predictions for each species (Figure 3). Model-performance metrics were calculated and used to evaluate the performance of each model-prediction.

### 2.7. Model Choice Evaluation

A multiple factor multivariate analysis of variance (MANOVA) was carried out to investigate the effect of TD, P, DR, and MT on five metrics commonly used to measure model performance. Kappa [48] developed by Cohen [49], area under the receiver operating characteristic curve (AUC) [50], sensitivity [48,51], specificity [48,51], and cross-validation error [52] given by the root mean square error (RMSE) of the variations in cross-validation iterations. MANOVA was used to obtain a statistically informed decision regarding which performance measure to use for evaluating factorial design results. Mahalanobis distances [53] derived between each performance score and the group (P, DR and MT) centroids were compared using the Chi-square (*x*^2^) distribution and plotted on a q–q plot [54], to determine multivariate normality of the model’s performance scores. The majority of the data conformed to the expected χ^2^ value with a few outliers. According to Box’s M test [55], the equality of variance assumption was fulfilled for P and MT, but not for TD and DR. We proceeded with the parametric MANOVA test considering the fact that none of the largest standard deviations within a group (factors) were more than four times larger than the smallest standard deviations within the same group, which suggested that MANOVA will still be robust [56]. As a cautionary measure, a conservative α value for the variables that failed the Box’s M test was used while carrying out the follow-up post-hoc group comparisons (i.e., α = 0.025 instead of the usual α = 0.05). Canonical correlation analysis was used to determine model performance measures that best described the effects of the modeling components. The standardized coefficients of the canonical correlation analysis were used to select the best model performance measures. The effects of predictor data type, dimension reduction, and model type on individual model performance indices were analyzed using single factor analysis of variance (ANOVA) with Tukey’s honestly significant difference (HSD) post-hoc test. The multivariate statistical analysis was carried out in R [42] statistical software version 3.0.2, and with packages agricolae [57], candisc [58], CCA [59], ggplot2 [60], heplots [61], hier.part [62], and multcomp [63]. The model with the maximum score for the chosen model performance measure based on the MANOVA analysis was ranked the best model, and the model with the lowest score was considered the worst model. For *V. vulgaris* predictions, external validation was undertaken as additional occurrence data was obtained after modeling was completed. Hierarchical partitioning [64,65] was carried out to quantify the independent contribution of modeling factors, TD, P, DR, and MT to mean Kappa and cross validation scores. 

Individual variables were ranked based on the frequency of their inclusion in the tested models. The method described by Dormann, et al. [9] was adapted for this purpose. The frequency of variable inclusion was calculated based on the number of times a variable was used by a model regardless of whether it was by a straight forward variable selection (RF) or by dimension reduction (PCA, and NLPCA).

Finally, the variability between the predictions across the 36 scenarios for each species were analyzed. Mean and standard deviation maps were also produced so that the spatial pattern of variability among the models can be easily visualized. The probability density of standard deviation by modeling components (P, DR, and MT) were plotted to investigate if any variation from the multivariate analysis of model performance scores was also reflected in the predicted data. 

## 3. Results

### 3.1. Multivariate Analysis of Variance (MANOVA)

The MANOVA results (Table 3) showed that all modeling components and their interactions had a significant effect on the linear combination of the five model performance scores (Kappa, AUC, sensitivity, specificity, and CV-error) with the exception of predictor choice (P). 

A follow up canonical correlation analysis was undertaken, and the first canonical variable accounted for 52.4% of model variance. The corresponding canonical correlation coefficient for the first variable was 0.9034 (Wilks λ = 0.015, F = 3.53, DF1 = 240, DF2 = 638, and *p* < 0.0001) showing strong linear correlation between the canonical factor loadings of performance measures and that of modeling components (Figure 4). Accordingly, the canonical coefficient of determination (R^2^ = 0.816) was also high, showing that the ratio of the explained variance out of the total variance was high. The standardized coefficients of the canonical correlation analysis showed that the Kappa score contributed to most of the variance of the first canonical variable (79.9%), and cross-validation error contributed to most of the second canonical variable (62.7%). Accordingly, model Kappa scores and cross validation error were selected for all subsequent comparisons and analyses (complete results of the canonical analyses are given in Appendix A). 

### 3.2. Quantifying the Variance Contribution of Modeling Factors

Individual follow-up ANOVA’s were performed for Kappa and cross-validation error scores, and the results largely agreed with the MANOVA analysis. The statistics for Kappa scores were as follows. All main effects were statistically significant (ANOVA test, SS > 0.24, η^2^ > 0.12, and *p* 0.000–0.002), with the exception of predictor choice (SS = 0.007, η^2^ = 0.003, and *p* = 0.764). Statistics for the interactions were also significant and comparable with main effects (ANOVA test, SS 0.17–0.52, η^2^ 0.03–0.05, and *p* 0.004–0.01). 

The hierarchical partitioning analysis identified TD as a source of the largest variation both in Kappa scores and in model cross-validation errors (54.8% and 47.5%, respectively), followed by MT, which accounted for 38.1% and 43.8% of the variation in Kappa and CV error scores, respectively. DR accounted for 6.8% in Kappa score variation and 8.6% in cross validation error variation, and P scored 0.2% and 0.1% for Kappa and cross validation error variation, respectively. 

### 3.3. Species Level Analysis of Variance

The species level model performance analysis showed that for *A. albopictus*, DR (ANOVA, SS = 0.118, df = 2, and *p* = 0.004), MT (SS = 0.689, df = 3, and *p* = <0.0001), and their interaction (SS = 0.259, df = 6, *p* = 0.001) had a significant effect on model Kappa scores. For *A. gracilipes*, only P had a significant effect on model performance scores (ANOVA, SS = 0.132, df = 2, and *p* = 0.004). However, pairwise comparisons of P and DR combinations (Tukey’s test, HSD = 0.24, and *α* = 0.05) showed that PCA-based dimension reduction gave the lowest Kappa scores for *A. gracilipes* predictions. *D. v. virgifera* results were similar to *A. albopictus,* except that the interaction between DR and MT was not significant. P (ANOVA, SS = 0.212, df = 2, and *p* = 0.026) and MT (ANOVA, SS = 0.552, df = 3, and *p* = 0.001) in *T. pityocampa,* had a significant effect on model performance. Finally, for *V. vulgaris*, only MT had a significant effect (ANOVA, SS = 0.128, df = 3, and *p* < 0.0001). 

### 3.4. Modeling Components

#### 3.4.1. Species Data

*A. albopictus* and *V. vulgaris* had the highest mean Kappa scores, suggesting that the highest prediction accuracy ranks were associated with species that had the largest presence data records (*A. albopictus*, *V. vulgaris*). Presence data prevalence was consistently associated with high prediction accuracy when compared among the five species. For example, *A. gracilipes* which had higher presence records than *D. v. virgifera* and *T. pityocampa,* had higher Kappa scores and lower CV error than both species. 

#### 3.4.2. Predictor Choice/Variable Selection

The variables mainly included across the different modeling combinations were annual precipitation (mm), precipitation of wettest quarter (mm), precipitation of driest quarter (mm), precipitation of warmest quarter (mm), mean temperature of wettest quarter (°C), precipitation of coldest quarter (mm), and annual mean temperature (°C). The complete rank is given in Appendix A.

#### 3.4.3. Dimension Reduction

DR interacted significantly with TD, where its effect on both Kappa and cross-validation error scores varied between species datasets. NLPCA generally outperformed both PCA and RF for all species except *T. pityocampa*, where the Kappa score from RF was slightly higher (see Appendix A for mean Kappa scores of all TD-DR-MT combinations). This is especially true for the two species *A. albopictus* and *V. vulgaris* that had a large number of presence point records covering large environmental and geographical areas. There was a difference in Kappa scores due to DR for some of the species. For example, Kappa value increased in magnitude by0.25 for the *D. v. virgifera* distribution model when using NLPCA compared with RF. RF had a higher Kappa and lower cross-validation error values compared to PCA for *T. pityocampa* and *A. albopictus*, while PCA performed better than RF for *D. v. virgifera* and *A. gracilipes*. The mean Kappa scores for RF and PCA were very similar for *V. vulgaris*. The generally poor performance of PCA reported by Dormann, et al. [9] was also observed in this study. With regard to the interaction with model types, there were no clear trends except for the LOGR model and PCA combinations, which consistently gave poorer model performance scores. 

#### 3.4.4. Model Type

Model type effect trend was consistent throughout all combinations of factors. The SVM model consistently outperformed the other three models (Appendix A). SVM and CART models were consistently ranked with the highest Kappa score and lowest CV-error groups. LOGR had a generally low Kappa and high CV-error scores throughout the factorial combinations. There was only one instance where LOGR scored better than QDA and CART for *A. gracilipes* within the group of models using the RF variable selection method. 

### 3.5. Model Ranking

Kappa score was used to rank model performance for the five species distribution predictions. Discriminating models that had similar Kappa scores was done by ranking them according to the cross-validation error values (Figure 5). The best and worst P-DR-MT combinations from the 36 scenarios for each species are given in Table 4. Worst prediction in this context does not imply that the reported dimension reduction or model types are not generally suited for the particular species, rather the recommendation is specific to the environmental data, presence records, and spatial extent used in this study. 

There were a number of interesting results where certain combinations did worse, despite belonging to a species with high presence prevalence. For example, the worst overall kappa score belonged to a prediction based on PCA-transformed data fitted by a logistic regression model for *A. albopictus*. Despite most of the predictions for *A. albopictus* being highly ranked according to Kappa scores (five out of the top 10 ranks out of 60 combinations; Appendix A), this particular prediction came last (60th) with a Kappa score of 0.34. When random forest variable selection was used instead of PCA, prediction for the same model (logistic regression) and species (*A. albopictus*) scored a Kappa = 0.72, which was ranked at 42 (Appendix A). 

Comparison between TD-DR-MT combinations (Appendix A) showed that model type could make a difference in prediction accuracy for some presence data, especially when the presence/pseudo-absence data were less reliable. For example, there were no statistically significant differences between Kappa scores for LOGR, QDA, CART, and SVM predictions for *V. vulgaris*. On the other hand, there was a statistically significant difference between Kappa scores of LOGR/QDA and SVM/CART for *T. pityocampa*, where the machine learning methods handled the low presence data prevalence better.

### 3.6. Species Distribution Predictions and Uncertainty

Model predictions were not examined until all the model performance score analyses were finalized. Once the best and worst modeling component combinations for all species were identified (Table 4), the corresponding predictions were examined. Most predictions were from the top five best models identified areas, which were well described as native or introduced geographical ranges of the five species studied (Appendix A). 

However, in all cases, the kappa scores of the best P-DR-MT combinations selected for the five species were not significantly different from the second best combinations. In some cases, the difference between the Kappa scores was not significant for the first 5–6 combinations. All of the worst modeling component combinations for the five species, however, had significantly lower kappa scores from the second worst modeling component combinations for the respective species. The second best modeling component combination (Figure 6C) for *A. gracilipes*, with the same dataset but a different dimension reduction method from the best combination (Figure 6B) with the second highest Kappa value, is shown in Figure 6. Clearly, the prediction of the second best Kappa model was similar to the best Kappa model. Assessing the probabilistic predictions for the five species showed that three models with the highest kappa scores, may have over-predicted the geographic ranges of the species under study (Appendix A). No additional validation data were available, thus assessment was done visually and only extreme predicted areas known to be outside the biological tolerance of the species were considered as over-predictions. 

The availability of various distribution prediction scenarios based on the different modeling component combinations allowed for the generation of standard deviation maps that could be used as uncertainty measures for predictions. The multi-scenario mean prediction and the associated prediction standard deviation maps for *A. albopictus* are given in Figure 7A,B, respectively (Appendix A shows data for the other species). 

The probability density for the standard deviation of model predictions given by the different modeling components is shown in Figure 8D for *A. albopictus* and Appendix A for the other species. 

## 4. Discussion

According to the MANOVA results, the effects of major modeling components such as model type, dimension-reduction, species dataset, and predictor dataset on model performance are in accordance with previous studies that investigated sources of uncertainty in SDM predictions [5,9,10,66,67,68]. Similarly, the per-species univariate analysis of variance conformed to the general trend with regard to the modeling components order of significance except for one case. For *A. gracilipes*, only predictor data had a significant effect on model performance, unlike the other four species where model type had the largest effect. This specific case shows that it is possible to have exceptions to the established trend requiring a case-by-case investigation to identify the most important modeling component whenever the modeling scenario changes. Change in modeling scenario here refers to change in occurrence data, species, predictor data, models, and any data pre-processing methods used. In this study, the *A. gracilipes* presence locations clearly clustered in the environmental feature space. Species with a limited environmental niche can be mapped using less complex models and limited variables, requiring a less complex modeling scenario (Reference [19], and references therein).

High presence data prevalence was associated with high model Kappa scores in all cases, however, prediction accuracy does not necessarily follow the size of the presence dataset as reported by Elith, et al. [5], based on their factorial study involving 226 species and 17 SDMs. Therefore, we drew no definitive conclusion from the strong correlation between presence data prevalence and high Kappa scores.

Predictor datasets: According to the frequency analysis for selected variables, more individual variables common to P1, P2, and P3 datasets were consistently included in the models than variables unique to P2 and P3 datasets (ranked 9th or higher out of the total rank of 18). The second most included set of variables were unique to dataset P2 and P3 (ranked 9th or higher out of the total rank of 18), therefore it is recommended to use the P2 dataset [31], which includes all the top half of the ranked list of predictors unless the modeler has good evidence that the target species distribution can be adequately described by temperature and precipitation derived variables alone (Appendix A). Elevation was the only variable unique to the P3 dataset that was consistently selected. The other three variables: slope, aspect, and hill-shade were only included in three models. Therefore, it seems that these three topographical variables may be more useful for higher resolution data at local scales than global or regional scale studies where elevation data can be used as a proxy for those variables. 

Dimension reduction method: Nonlinear principal component analysis (NLPCA) appeared to perform well based on comparisons of Kappa and cross-validation error. The most important information concerning dimension reduction methods, however, was obtained from assessing the actual predictions rather than the statistical pairwise comparisons of model Kappa scores. 

Three of the five models with the highest Kappa score selected for the five species had used the h-NLPCA (DR3) as a dimension reduction method, and when their corresponding predictions were assessed it was apparent that all three models had over-predicted. Deciding whether a model has over-predicted is not an easy task unless there is additional external validation data that covers areas beyond what is covered by the training/test data. In the case of the three models selected for *A. albopictus*, *D. v. virgifera*, and *V. vulgaris*, determining whether they were over-predicting was straightforward because the areas that were incorrectly predicted were in areas where the environmental conditions were outside the known biological tolerance of these three species (Appendix A). For example, some of these areas were Greenland for *A. albopictus*, Sahara, and the Middle East for *D. v. virgifera* and *V. vulgaris*. 

Exploration of relative locations for the presence points of the three species and their corresponding pseudo-absences points selected from the h-NLPCA (DR3) transformed data, revealed a possible reason for the over-prediction. The pseudo-absences selected from the h-NLPCA transformed datasets were highly discriminated from the presence points, and highly localized in the predictor feature space. While high discrimination between presences and pseudo-absences in the feature space can be desirable for a clear characterization of suitable and unsuitable habitats, the highly localized pseudo-absence points are problematic, as that means less information regarding what constitutes unsuitable conditions in the environmental space is available. The high Kappa scores for the h-NLPCA dimension reduction method indicate that h-NLPCA may be a useful tool in species distribution modeling, provided that the tendency for models to over-fit on h-NLPCA-transformed data can be addressed through incurring a training gain penalty [69] or a regularization scheme [70]. 

Model type: Machine-learning methods were consistently highly ranked in performance for species with high data prevalence covering large portions of environmental space, while QDA performed better with species that had low prevalence—where occurrences occupied a localized area in environmental space. Similar results were reported by Segurado and Araújo [71], in their study that evaluated commonly used species distribution models. This result leads to the conclusion that each species should be treated individually, and that model selection should be solely based on the occurrence data used and not on recommendations from other studies that have used very different presence data and environmental variables. Several caveats should be noted. Some models maybe positively or negatively affected by the pseudo-absences to presences ratio during model training. For example, regression-modeling techniques perform better when disproportionately more pseudo-absences (zeros) are used for fitting [72,73,74]. In our study, we kept the number of pseudo-absences equal to presences, to ensure that factors not measured for effect are kept constant for an unbiased comparison of model outcomes. In addition, we believe that the LOGR model may, on average, have performed worse than other models because different regularization options were not tested [73]. Our objective was to test the models in their commonly used formats, while individually parameterizing each model was outside of the scope of this study. 

Performance measures: Indices based on the confusion matrix are the most used methods for model performance measurement in species distribution modeling [51]. These methods have been widely and successfully used in other disciplines, especially in clinical studies, long before they were adapted for ecological modeling [75]. However, methods based on the confusion matrix are not always sufficient for model validation in the ecological context, in cases where minimal training data is used for species distribution predictions that cover a much greater geographical area than the training data [75,76,77]. This training data/prediction imbalance is especially pronounced when SDMs are used for regional or global studies. An example in this study is that none of the models with the highest kappa scores were statistically different from the second or third highest score models. This equivalence effectively means that different modeling factor combinations used in the second or third highest kappa score models could give similar or sometimes better (in case of models that over-fit or extrapolate [78,79]) predictions than the actual highest Kappa score model. There is therefore a need to devise additional model prediction evaluation tools beyond those generated from a confusion matrix.

## 5. Conclusions

This study shows that the predictive performance of different model types depends mainly on data pre-processing, in other words, on pseudo-absence dataset development, dimension-reduction method, species- and predictor-dataset selection. Thus, performance comparisons among different model types cannot be applied unless all data pre-processing factors are kept constant. The model-type comparisons applied for the same species and predictor datasets and under the same dimension-reduction method showed that machine learning models (mainly SVM) outperform other model types, with the exception that less complex statistical models can perform well for species with a limited environmental niche. h-NLPCA was also a highly ranked dimension-reduction method. Variation in SDM predictions does not always necessarily occur due to reasons inherent in the model types used. Thus, it is possible to achieve better SDM predictions using appropriate modeling components such as predictors and dimension reduction methods that fit the available data. The variation in species distribution prediction was observed when we used different modeling components coupled with the low discriminatory power of evaluation methods among top performing models, supports two recommendations for modeling species distributions to increase prediction certainty: 1) where possible explore the use of multi-scenario modeling frameworks where a variation of environmental variables, dimension-reduction methods, and model types are tested in a standardized manner before picking the best modeling combination; and 2) use a second model performance measure to allow discriminating model combinations that might have identical or very similar scores based on the first model performance measure. Performing species distribution predictions in a multi-scenario modeling framework can also provide some measure of uncertainty through assessment of model consensus between predictions made for the same species, using different modeling components. When resources available for control or eradication of an invading insect population are limited, the design of sampling protocols for detection may be focused on predicted locations that are accompanied by high model consensus or low-uncertainty hotspots.

## Figures and Tables

**Figure 1 insects-10-00065-f001:**
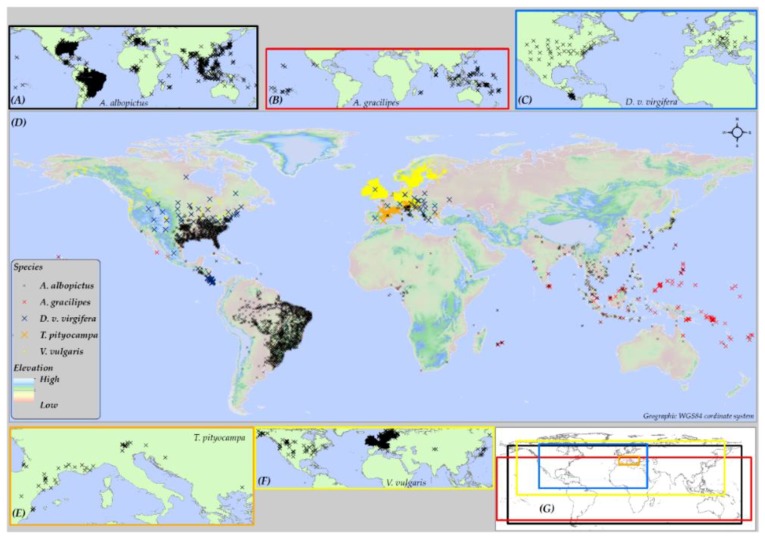
Maps showing the global occurrence of the five species used in this study. Inset maps show the global distribution of (**A**) *Aedes albopictus*; (**B**) *Anoplopis gracilipes*; (**C**) *D. v. virgifera*; (**E**) *Thaumetopoea pityocampa*; and (**F**) *Vespula vulgaris*. (**G**) The geographic extent of each species; and main map (**D**) shows the extent of occurrence of all five species with presence points overlaid on a global elevation model.

**Figure 2 insects-10-00065-f002:**
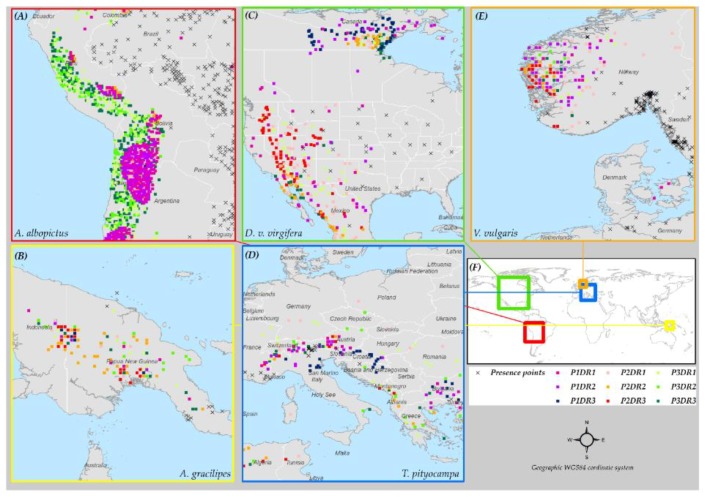
Subsets of the global study area with different sets of pseudo-absence points shown by the species, predictor data, and dimension reduction method. The nine sets of pseudo-absences generated based on the different combinations of the three predictor datasets and three dimension reduction methods are shown for *A. albopictus* (**A**), *A. gracilipes* (**B**), *D. v. virgifera* (**C**), *T. pityocampa* (**D**), and *V. vulgaris* (**E**). The extents of the sub-set maps (**A**–**E**) are shown on the global map (**F**).

**Figure 3 insects-10-00065-f003:**
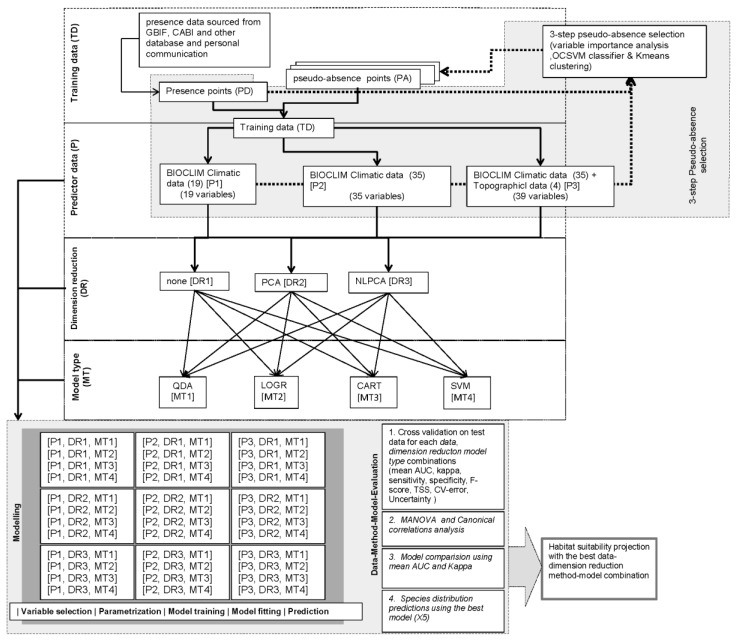
Conceptual model showing factorial research design. The study was carried out using a 3 × 5 × 3 × 4 factorial design. The design incorporated three types of predictor datasets, occurrence data for five species, three types of collinearity reduction methods, and four types of models that utilize different modeling techniques.

**Figure 4 insects-10-00065-f004:**
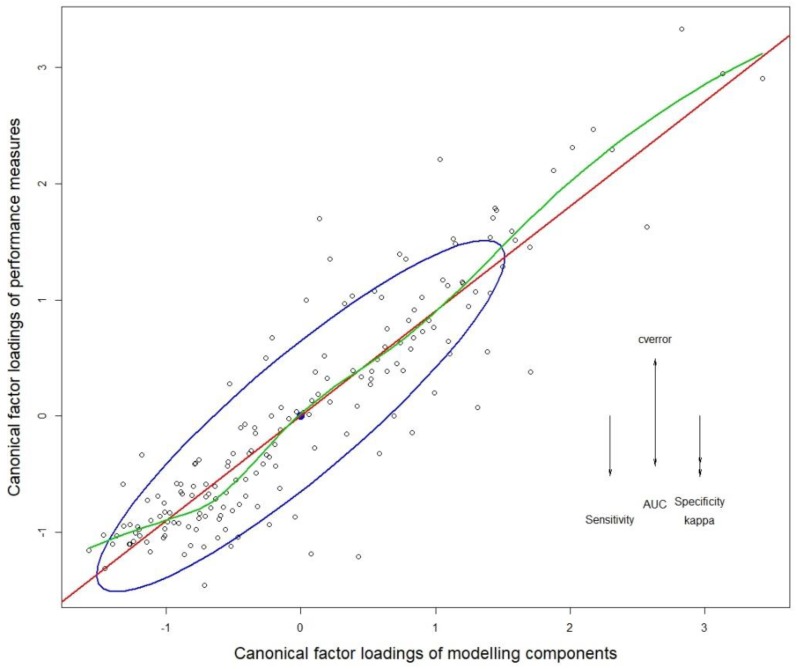
Structure correlations (canonical factor loadings) for the first canonical dimension. Arrows show the vector direction of variables that correspond to the canonical component on the y-axis. The corresponding variables for the x-axis (combinations of modeling components) were not labeled to avoid overcrowding the graph. The red line indicates the linear regression line. The blue ellipse (data ellipse) shows 68% of the data points (approx. one standard deviation and their centroid (filled black dot) in relation to the linear regression line. The green line shows the locally weighted scatterplot smoothing (LOWESS) fit.

**Figure 5 insects-10-00065-f005:**
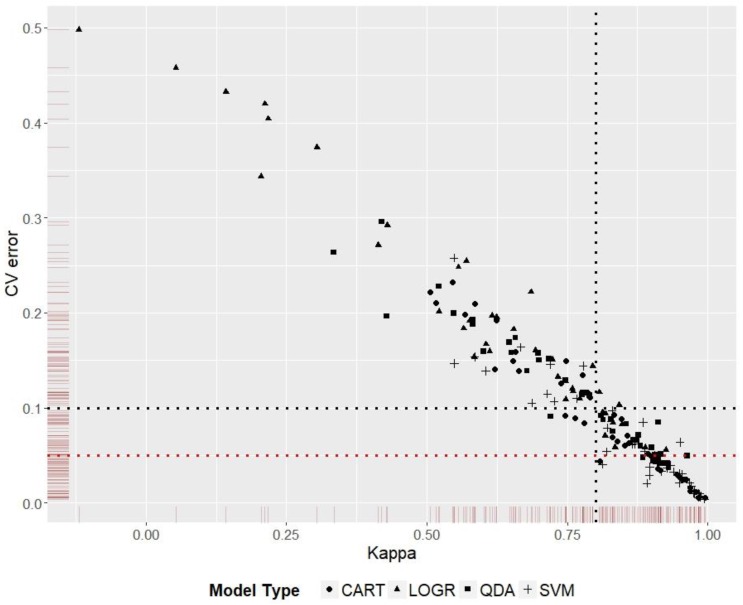
Model Kappa scores plotted against cross-validation error scores. Models to the right of the vertical black dotted line have a Kappa score ≥0.8; models below the horizontal black dotted line have a cross validation error ≤0.1, and models below the horizontal red dotted line have a cross validation error ≤0.05. The graph shows the advantage of using a second performance score to discriminate between models with similar scores on the first performance measure.

**Figure 6 insects-10-00065-f006:**
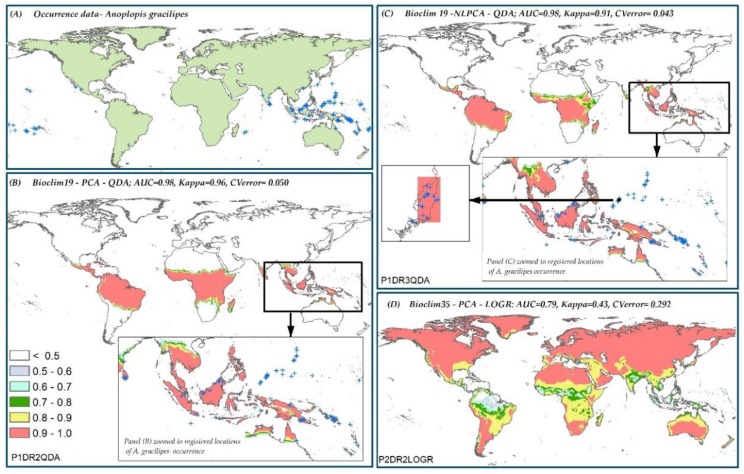
Predicted probability of presence for *A. gracilipes*. (**A**) Occurrence data, (**B**) the best model combination for *A. gracilipes*, (**C**) the second best model combination for *A. gracilipes*, and (**D**) the worst model combination for *A. gracilipes*.

**Figure 7 insects-10-00065-f007:**
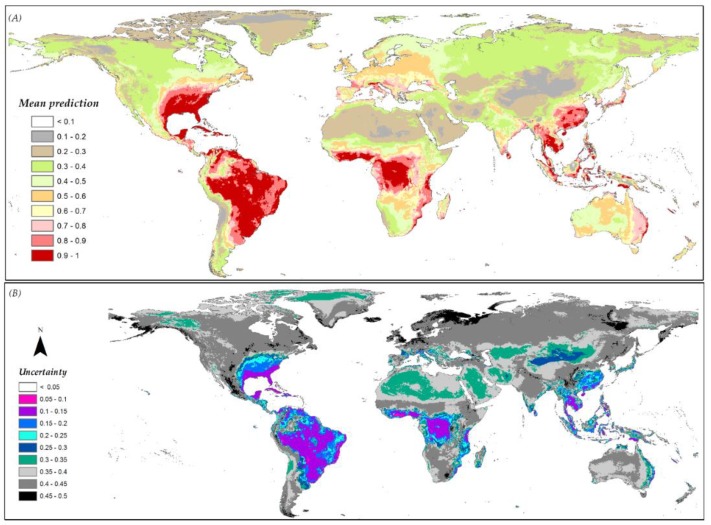
(**A**) Mean predicted presence across all scenarios for *A. albopictus*; (**B**) the associated uncertainty around the mean prediction within the multi-scenario modeling framework. Grey shades show higher uncertainty whereas purplish-bluish shades show lower uncertainty in the form of low SD among replicates.

**Figure 8 insects-10-00065-f008:**
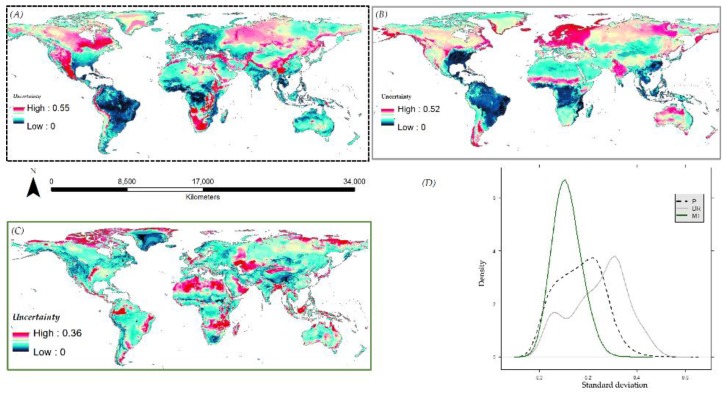
Spatial pattern of variability according to, (**A**) predictor data—P, (**B**) dimension reduction—DR, (**C**) model type—MT, and (**D**) the probability density of predicted presences according to the P, DR, and MT for *A. albopictus*.

**Table 1 insects-10-00065-t001:** Variables included in the three-predictor datasets used in this study.

Variable	Variable Name	Dataset
01	Annual mean temperature (°C)	P1, P2, P3
02	Mean diurnal temperature range (mean(period max-min)) (°C)	P1, P2, P3
03	Isothermality (Bio02 ÷ Bio07)	P1, P2, P3
04	Temperature seasonality (C of V)	P1, P2, P3
05	Max temperature of warmest week (°C)	P1, P2, P3
06	Min temperature of coldest week (°C)	P1, P2, P3
07	Temperature annual range (Bio05-Bio06) (°C)	P1, P2, P3
08	Mean temperature of wettest quarter (°C)	P1, P2, P3
09	Mean temperature of driest quarter (°C)	P1, P2, P3
10	Mean temperature of warmest quarter (°C)	P1, P2, P3
11	Mean temperature of coldest quarter (°C)	P1, P2, P3
12	Annual precipitation (mm)	P1, P2, P3
13	Precipitation of wettest week (mm)	P1, P2, P3
14	Precipitation of driest week (mm)	P1, P2, P3
15	Precipitation seasonality (C of V)	P1, P2, P3
16	Precipitation of wettest quarter (mm)	P1, P2, P3
17	Precipitation of driest quarter (mm)	P1, P2, P3
18	Precipitation of warmest quarter (mm)	P1, P2, P3
19	Precipitation of coldest quarter (mm)	P1, P2, P3
20	Annual mean radiation (W m^−2^)	P2, P3
21	Highest weekly radiation (W m^−2^)	P2, P3
22	Lowest weekly radiation (W m^−2^)	P2, P3
23	Radiation seasonality (C of V)	P2, P3
24	Radiation of wettest quarter (W m^−2^)	P2, P3
25	Radiation of driest quarter (W m^−2^)	P2, P3
26	Radiation of warmest quarter (W m^−2^)	P2, P3
27	Radiation of coldest quarter (W m^−2^)	P2, P3
28	Annual mean moisture index	P2, P3
29	Highest weekly moisture index	P2, P3
30	Lowest weekly moisture index	P2, P3
31	Moisture index seasonality (C of V)	P2, P3
32	Mean moisture index of wettest quarter	P2, P3
33	Mean moisture index of driest quarter	P2, P3
34	Mean moisture index of warmest quarter	P2, P3
35	Mean moisture index of coldest quarter	P2, P3
36	Elevation (m)	P3
37	Slope (deg)	P3
38	Aspect (deg)	P3
39	Hillshade	P3

**Table 2 insects-10-00065-t002:** Number of presence points * and distances used to limit background extent before pseudo-absence selection, for the three types of predictor datasets used to model the global distribution of the five species in this study.

No.	Species	Predictor	Distance (km)
1	*Aedes albopictus* (3029/2928)	BIOCLIM19	350
2	*Aedes albopictus* (3029/2928)	BIOCLIM35	300
3	*Aedes albopictus* (3029/2928)	BIOCLIM35+T4	600
4	*Anoplopis gracilipes* (385/101)	BIOCLIM19	550
5	*Anoplopis gracilipes* (385/101)	BIOCLIM35	500
6	*Anoplopis gracilipes* (385/101)	BIOCLIM35+T4	400
7	*Diabrotica v. virgifera* (449/84)	BIOCLIM19	2000
8	*Diabrotica v. virgifera* (449/84)	BIOCLIM35	800
9	*Diabrotica v. virgifera* (449/84)	BIOCLIM35+T4	800
10	*Thaumetopoea pityocampa* (67/33)	BIOCLIM19	300
11	*Thaumetopoea pityocampa* (67/33)	BIOCLIM35	1300
12	*Thaumetopoea pityocampa* (67/33)	BIOCLIM35+T4	800
13	*Vespula vulgaris* (10,048/920)	BIOCLIM19	550
14	*Vespula vulgaris* (10,048/920)	BIOCLIM35	300
15	*Vespula vulgaris* (10,048/920)	BIOCLIM35+T4	700

* Numbers next to each species name show available presence points followed by spatially unique points with respect to the environmental predictor dataset resolution. Another two sets of the datasets listed above were generated according to the listed background binding distances for predictor data transformed using PCA and NLPCA making 45 training/test datasets in total. BIOCLIM19 contains 19 precipitation- and temperature-based variables, BIOCLIM35 contains variables in BIOCLIM19 plus 26 radiation- and soil moisture-derived variables, BIOCLIM35+T4 contains variables in BIOCLIM35 plus 4 variables derived from topographic data.

**Table 3 insects-10-00065-t003:** The multiple factor multivariate analysis of variance (MANOVA) results table showing the effects of the various modeling components model performance.

Modeling Components	Pillai’s Trace	η^2^ (%)	F	Df	*P* ^#^
Model type	0.79	26.22	9.24	3	<0.001 ***
Dimension reduction	0.42	21.01	6.86	2	<0.001 ***
Species data	0.81	20.32	6.68	4	<0.001 ***
Predictor	0.11	5.50	1.50	2	0.138 ^ns^
Species data x Predictor	0.68	13.51	2.58	8	<0.001 ***
Species data x Dimension reduction	0.58	11.65	2.18	8	<0.001 ***
Predictor x Dimension reduction	0.49	12.37	3.70	4	<0.001 ***
Species data x Predictor x Dim. Red.	0.95	18.98	1.93	16	<0.001 ***
Residuals		26.22		132	

^#^ Signif. *P* codes: 0 < *** ≤ 0.001 < ** ≤ 0.01 < * ≤ 0.05 ≤ 0.1 < ns ≤ 1.

**Table 4 insects-10-00065-t004:** Best and worst component combinations for the five species modeled in this study. For abbreviations, please refer to Figure 3.

Species	Best	Kappa	CVerror	Worst ^#^	Kappa	CVerror
*A. albopictus*	P_1_DR_3_SVM *	0.99	0.006	P_1_DR_2_LOGR	0.14	0.433
*A. gracilipes*	P_1_DR_2_QDA *	0.96	0.050	P_2_DR_2_LOGR	0.43	0.292
*D. v. virgifera*	P_1_DR_3_SVM *	0.98	0.006	P_2_DR_2_LOGR	0.21	0.344
*T. pityocampa*	P_2_DR_2_SVM *	0.88	0.009	P_3_DR_3_LOGR	−0.12	0.498
*V. vulgaris 1*	P_1_DR_3_SVM *	0.99	0.004	P_1_DR_3_LOGR	0.56	0.248
*V. vulgaris 2*	P_1_DR_3_CART *	0.99	0.005			

* Combinations are the best based on their high Kappa and low CVerror, but are not significantly different from the second best combination. ^#^ All model combinations identified as “worst” for a species had a significantly lower score than the second worst models. CVerror = cross-validation error. For *V. vulgaris,* additional presence data was obtained from the Landcare Research Centre (Appendix A). External validation of the selected model for *V. vulgaris* using the New Zealand *V. vulgaris* presence data showed that 91% of the occurrence sites were correctly predicted by the selected model. Two combinations were selected for *V. vulgaris* as they had the same Kappa score, CV error was used to select from the two equivalent Kappa score models. P and DR indicate the predictor data and dimension reduction method used along with the models selected as best or worst models.

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
