# Peer review of "Multi-Scenario Species Distribution Modeling"

_insects, 2019, doi:10.3390/insects10030065_

Round 1

Reviewer 1 Report

The manuscript presents a comparison of correlative modeling approaches for species distribution models. In particular, the authors compare 4 correlative model types and the choice of predictor variables used to investigate the effect of modelers choice. The study provides a valuable illustration of the importance of modeler’s choice of predictors and data reduction methods. The study is valuable for both researchers and managers interested in the practical use of correlative models. The manuscript is generally well written, but the organization of a few sections makes the manuscript a little difficult to read. With a few editorial changes this manuscript should prove to be a valuable contribution to the literature. Please see editorial comments below.

Editorial comments:

Abstract: The abstract could use a few specific results from the study.

Introduction:

Lines 64-72: This section of the manuscript should be expanded to provide the reader a clear road map for the rest of the article. The material and methods section is fairly dense and having a smoother introduction to the articles main points would be very useful.

Materials and Methods:

Please reconsider the order in which sections are placed. Some of the methods presented appear early in the methods section and are explained later.

Lines 80-97: This section is very important to follow the manuscript and should be expanded and possibly consider moving the section to a later point in the methods section. Many of the components stated in this section, the species, predictor variables, data reduction methods, pseudo-absence absence methods, etc. are well described later in the manuscript. After reading the methods this section was very clear, but upon first reading it took me a little while to get through. For example, at first I assumed the species and variables used would be simulated/imaginary species. It isn’t until later in the manuscript that the fact that the real species used are mentioned. A potential location could be just prior to the “model types” section.

Line 112: P2 datasets are not described above.

Line 150: PCA is described in lines 192-196. Given the presence of a section about data reduction methods and the appendices, it is assumed that some of the intended audience may be unfamiliar with these methods. Therefore, the section on data reduction methods should be placed before they are stated.

Line 209: Please remove “more or less”

Lines 207-233: Some of these methods (particularly logistic regression) can have issues with excess zeroes, which could be an issue here. Please state whether this was an issue or not. 

Results:

Line 295 and later: Please consider using a statement about how to interpret eta-squared prior to discussing it (e.g. Cohen’s guideline 0.01 small, 0.06 medium, and 0.14 large).  

Discussion:

Lines 451-453: The statements “Such a distinct grouping…. This made predictor data choice more important than model type in this case”, did not make sense to me. It would appear that there was no effect of either model type or predictors. The maps also didn’t seem to fit that well with the predicted data, I couldn’t really see the colors in the islands well. A close up of the locations may be useful.   

Line 520: The statement, “Therefore, the question should not be that the predictions varied, it is more of which one of the predictions are appropriate for a given species represented by a given occurrence dataset”, need to be edited. I understand the point being made and it needs to be made. However, the first component is not a question. 

Author Response

Response to comments of Reviewer 1

Comments and Suggestions for Authors

The manuscript presents a comparison of correlative modeling approaches for species distribution models. In particular, the authors compare 4 correlative model types and the choice of predictor variables used to investigate the effect of modelers choice. The study provides a valuable illustration of the importance of modeler’s choice of predictors and data reduction methods. The study is valuable for both researchers and managers interested in the practical use of correlative models. The manuscript is generally well written, but the organization of a few sections makes the manuscript a little difficult to read. With a few editorial changes this manuscript should prove to be a valuable contribution to the literature. Please see editorial comments below.

>> Thank you, we have attempted to make alterations as per your review to make the MS more organized and focused. We appreciated the guidance we got from your review in terms of re-structuring the MS. We are also grateful for the key technical and theoretical issues you raised; we believe the MS is better understandable after those alterations. I hope you find the updated version better organized.

 Editorial comments:

Abstract: The abstract could use a few specific results from the study.

>> In addition to our two major results, 1) That model type is the major source of model discrepancy and 2) that changes in modelling components such as species data richness, dimension reduction and variable type can affect SDM prediction accuracy, we have added our observation that widely used model performance methods used currently in SDM model evaluation are not sensitive enough to discriminate closely performing models and that research to determine a more robust evaluation techniques is needed. 

Introduction:

Lines 64-72: This section of the manuscript should be expanded to provide the reader a clear road map for the rest of the article. The material and methods section is fairly dense and having a smoother introduction to the articles main points would be very useful.

>> We have instead expanded on earlier paragraphs to give better background on the types of modelling factors we will be testing and kept the last paragraph as a summary of what we plan to do with minor edits.

>> We expanded the introduction by giving background information on why we each of the modelling components we selected to test are important in determining predictive accuracy of SDMs. The new edits are found in:

>> We think the reason why differences in model type (model algorithms) is a significant component in affecting SDM predictive accuracy is sufficiently given in L41-52 of the updated MS.

>> Expanded background on why predictor choice is an important factor now given in L54-74 of the updated MS

>> Expanded the rationale for testing selected dimension reduction methods on L83 –L86 of the updated MS.

Materials and Methods:

Please reconsider the order in which sections are placed. Some of the methods presented appear early in the methods section and are explained later.

 Lines 80-97: This section is very important to follow the manuscript and should be expanded and possibly consider moving the section to a later point in the methods section. Many of the components stated in this section, the species, predictor variables, data reduction methods, pseudo-absence absence methods, etc. are well described later in the manuscript. After reading the methods this section was very clear, but upon first reading it took me a little while to get through. For example, at first I assumed the species and variables used would be simulated/imaginary species. It isn’t until later in the manuscript that the fact that the real species used are mentioned. A potential location could be just prior to the “model types” section.

>> We agree, thank you. This is now moved after the “model types” and “Model choice evaluation” sections, as the “research and design” section and the figure therein summarizes the different model types and the model evaluation techniques described in those two sections respectively.  Current location L277-287 in the updated MS.

 Line 112: P2 datasets are not described above.

 >> We described P2 on line 107 (L107 of the updated MS), but we agree that putting the “described above” phrase might lead the reader to believe it was described in a different section rather than two lines up, so we removed this phrase.

Line 150: PCA is described in lines 192-196. Given the presence of a section about data reduction methods and the appendices, it is assumed that some of the intended audience may be unfamiliar with these methods. Therefore, the section on data reduction methods should be placed before they are stated.

>> We have moved the “Dimension reduction” section to L131-149 of the updated MS to ahead of the “species data” section.

 Line 209: Please remove “more or less”

>> Done

 Lines 207-233: Some of these methods (particularly logistic regression) can have issues with excess zeroes, which could be an issue here. Please state whether this was an issue or not. 

>> In general, models may perform better with a presence-absence ratio that is custom fit for the modelling scenario [1]. We agree that logistic regression in particular may perform better when fitted with excessive pseudo-absences (zero labels). In our research design equal number of presences to pseudo-absences were used throughout the factorial experiment to keep the system constant for a proper comparison of model outcomes. Even if it is not the absolutely ideal set up for each model, we considered it important to keep as many constant factors as possible beyond the identified factors being tested for effect. Additional to the pseudo-absence ratio issue, we think that the LOGR model may have performed worse than other models in this study as regularization options were not tested [2] , Detailed individual parametrizing of each modelling methods was outside of the scope of this study. But we have now mentioned both caveats in L520- L528 of the updated MS.   

 Results:

Line 295 and later: Please consider using a statement about how to interpret eta-squared prior to discussing it (e.g. Cohen’s guideline 0.01 small, 0.06 medium, and 0.14 large).  

>> Thank you for bringing this to our attention. We have now included p values so that readers can easily identify significant effects. We thought the significance values will suffice therefore we did not include guidance for eta^2 interpretation in the interest of simplification. However, we are open to including it if you believe it is still necessary to include that information.

 Discussion:

Lines 451-453: The statements “Such a distinct grouping…. This made predictor data choice more important than model type in this case”, did not make sense to me. It would appear that there was no effect of either model type or predictors. The maps also didn’t seem to fit that well with the predicted data, I couldn’t really see the colors in the islands well. A close up of the locations may be useful.   

>> We agree that there was not a significant effect from both MT and DR for A. gracilipes predictions.  We were trying to make the point that complex models and data pre-processing methods are not always needed to achieve good model performance. Now replaced with “Species that have limited environmental range can be mapped with less complex models and limited variables, requiring a less complex modeling scenario [3 and references therein]“ in L469 -470 of the updated MS.

>> We have provided a close-up of the Pacific Island areas where A. gracilipes occurs in Figure 6.

 Line 520: The statement, “Therefore, the question should not be that the predictions varied, it is more of which one of the predictions are appropriate for a given species represented by a given occurrence dataset”, need to be edited. I understand the point being made and it needs to be made. However, the first component is not a question. 

>> This is now replaced by new text, L550-556 in the updated MS.

Reference:

1.             Barbet-Massin, M.; Jiguet, F.; Albert, C.H.; Thuiller, W. Selecting pseudo-absences for species distribution models: how, where and how many? Methods in Ecology and Evolution 2012, 3, 327-338, doi:10.1111/j.2041-210X.2011.00172.x.

2.             Gastón, A.; García-Viñas, J.I. Modelling species distributions with penalised logistic regressions: A comparison with maximum entropy models. Ecological Modelling 2011, 222, 2037-2041, doi:https://doi.org/10.1016/j.ecolmodel.2011.04.015.

3.             Jiménez-Valverde, A.; Lobo, J.M.; Hortal, J. Not as good as they seem: the importance of concepts in species distribution modelling. Diversity and Distributions 2008, 14, 885-890, doi:10.1111/j.1472-4642.2008.00496.x.

Reviewer 2 Report

Overall, this is a very nice work that needs much reforming to be adequately presented.

See my attached file for details.

Author Response

Response to comments of Reviewer 2

Comments and Suggestions for Authors

Overall, this is a very nice work that needs much reforming to be adequately presented.

>> We would very much like to thank you for taking the time it needed to thoroughly go through our manuscript and rearrange the structure of the article going beyond the time an average review will take. We appreciate all the comments, suggestions and edits you forwarded to us and we have tried our best to edit our MS accordingly. We are also grateful for your guidance in making our statistical result presentations clearer. We believe the MS in a much better shape because of these recommendations. In cases where we felt what we meant to say was misunderstood due to lack of clarity of how the original MS was written, we have provided more explanation. Thank you again.

See my attached file for details.

This article compares the predictive accuracy of various species distribution models within various insect species datasets, predictor variables and dimension reduction methods. The whole work is very interesting and, as far as my expertise allows me to judge, the methods followed are well implemented and adequate BUT, the presentation of the methods, results, including images and tables needs much reforming to be informative to a reader who reads this article for the first time.

>> Thank you! We hope it is better structured now with the help of the reviewers’ comments.

It took me many hours of reading and reading again, and going up and down in the text to understand what exactly it was applied but, when I got the meaning, I realized that all this information could be described in a more easily understood manner and this cannot be undertaken in only one revision.

Therefore, for this first round, I paid much attention to the ‘adequate presentation’ of your work. I reviewed the article until the ‘Results’ section and after the authors apply the suggestions, I will further proceed to the ‘Discussion’ and ‘Conclusions’ and also check the ‘connection’ between the purpose of the study, the results and the discussion-conclusions (from a first look, they seem ok with some minor amendments). My current intension is to make the article as informative as it gets and ease the understanding from the very first read, because the authors did much work but in this version, this work is not well presented.

>> Thank you, and we have only edited the MS up to the end of the methods section (except in cases where we addressed sections to be moved to methods or comments from reviewer 1), as we wanted to follow your suggested 2-step review process.

My general suggestions:

(1) Please have someone check the English usage. I am not a native speaker and from what I saw, there are people at your team that could make such an editing.

>> The first authors takes responsibility for this. There was not enough time to look into the English usage by my co-author. We focused on getting it ready quickly for the special issue. We will go through it again after the second round.

(2) There are many parts in your text that do not fit the section they are currently in. Some sentences in the ‘Results’ section for example are appropriate for the ‘Materials and Methods’ or the ‘Discussion’ section. All methods should be described in the ‘Materials and Methods’. The ‘Results’ should include only your results without commenting on them. Leave the comments for the ‘Discussion’. And focus on your objectives set at the end of the ‘Introduction’ section.

>> This has been addressed in the updated MS.

(3) Be simple and precise. Check my detailed suggestions below. There is too much unnecessary ‘talking’ and this distracts the reader from what is being discussed/described.

>> Thank you for all the great suggestions. We hope that you will find the updated MS more focused.

(4) Check all figures and tables and explain in the caption every abbreviation, color codes etc. For example, Table S3 also includes a figure and I don’t see any ‘bar’ as described in the caption.

>> This has been updated throughout

Check my detailed suggestions and apply the same concept (of writing simple, precise sentences without unnecessary words and phrases) throughout the text. And in the next revision round, I think that this article will be way better in terms of adequately presenting and discussing the large amount of work that you have undertaken.

Congratulations for the work done and good luck with your revision.

>> Thank you! We have updated the MS according to the review, we hope you will find it better organized now.

ABSTRACT

Line 13. I don’t understand the ‘invasive’ part. And maybe it’s not necessary here. Why not ‘Correlative species distribution models (SDMs) are increasingly being used to predict suitable insect habitats’.

>> Updated according to the comment

Line 13. ‘… there is also increased criticism …’

>>Done

Line 16. ‘… the effects of various modelling components on …’

>>Done

Line 17. ‘The components-factors analyzed…’

>>Done

Line 15-18. An alternative to your sentence. ‘We undertook a factorial study to investigate the effect of various modelling components (species -training- datasets, predictor variables, dimension reduction methods, modelling algorithms) on the accuracy of SDM predictions with an aim to discern sources of discrepancy and uncertainty.’

>>Done

Line 18-20. This sentence is also not informative and could be removed.

>>Done

Line 21. ‘… the predictive accuracy’.

>> Here, we are referring to the accuracy of the final predictions (distribution maps), more importantly the point we wanted to make was about the variation in potential distribution maps from different SDMs.

We have made a change to “species distribution predictions” for clarity. What we would like to convey is that the variation in the distribution maps from different SDMs is mainly sourced from the type of model used.  We will be glad to change this in the next round if you still feel like it does not describe what we intend to describe.

Line 21-22. ‘It was additionally observed that a different combination …’

>>Done

INTRODUCTION

Line 34. Although the terms ‘species distribution models’ and ‘habitat suitability models’ are often used interchangeably, they do not mean the same thing so, just stick to the ‘species distribution models’ term.

>>Done

Line 34. Delete ‘of a number’.  

>>Done

Line 35. It is either ‘correlative models that use’ or ‘correlative modelling that uses’.

>>Done

Line 37. ‘Currently, the occurrence of discrepancies …’. Delete ‘the field of’.

>>Done

Line 38. You can avoid repetition by combining the sentences ‘… ecological modelling and the need for quantification of model uncertainty has been repeatedly …’.

>>Done

Line 41. Prediction accuracy or predictive accuracy? Use ‘variation’ instead of difference to avoid repetition.

>>Done

Lines 42-43. ‘have been reported’?

>>Done

Line 43. ‘Accurate’ or ‘suitable’ or ‘adequate’ instead of ‘good’? ‘for the prediction’? For the prediction of what? Maybe say ‘for predicting the distribution of rare species’.

>> Changed to “suitable for predicting the distribution of rare species”

Line 44. ‘… while complex models, such as …’.

>>Done

Line 47. ‘multidimensional’ instead of ‘high dimensional’?

>>We agree that is the correct descriptor. We changed to “multidimensional”

Lines 48-49. ‘model-prediction results’. ‘relative-occurrence areas’.

>>Done

Line 53-54. ‘further complicate the gradient’? Do you mean ‘further increase the divergence in the predictions between simple and complex models’?

>>Done

Line 55. ‘… because the representation of the interactions among environmental variables affects the predictive accuracy …’

>>Done

Line 64. It’s ‘sources of uncertainty’. Uncertainty is one. The same applies in line 65.

>>Done

Line 66. ‘… we investigated’.

>>Done

Line 69. You have already explored, so, ‘We also explored …’.

>>Done

Line 70. Either ‘multi-scenario modelling frameworks’ or ‘a multi-scenario modelling framework’.

>>Done

Line 71-72. Move these lines to line 66. The purpose of this study was to investigate the effect … We further explored whether it is possible … We finally explored the advantage …

>>Edited accordingly

MATERIALS AND METHODS

Lines 76-79. These lines are not informative at all. If your dataset is the global distribution of five insects, there is no point in telling now that your study area is the earth. Remove these lines, combine them with the next subsections and start with line 80.

>> We believe it is important to give study area extents explicitly in modelling papers as people who are searching for information on a given geographic area can quickly find this information. We have removed the portion that states, “All datasets used in this study cover major land masses with the exception of Antarctica” as the presence datasets actually cover various spatial extents. We have now added the additional information that the data for the different species cover different extents along with other space (extent) related metadata like resolution.  We are prepared to combine this section with the research design section if the rationale given here is still not convincing.

Research design and model conceptualization. Presence-only data for five insect species (TD) were used and sets of pseudo-absence (PA) data were developed for three predictor datasets (P) and three dimension-reduction (DR) methods (in total 5 TD x 3 P x 3 DR = 45 PA datasets) (Fig. 1). Within each PA dataset 80% of the data was used for model training and 20% for model evaluation. The PA datasets for each species were used to train and evaluate four different models-algorithms (MT). Each MT was used to predict the global distribution of each species based on the various P and DR combinations (in total 3 P x 3 DR x 4 MT = 36 different predictions for each species). Model-performance metrics were calculated and were used to evaluate the performance of each model-prediction.

>>  We have updated this section according to your edits. We have only made one alteration, since the training data used to train the models is a combination of presences and pseudo-absences, we now make it clear that the presence data was used to generate the pseudo-absence datasets, after which a training dataset that combines presence and pseudo-absence points for each species-P-DR combinations was prepared.

Predictor datasets. Remove (P, Abiotic)

Three predictor datasets were used (Table 1), (i) the BIOCLIM19 (P1) consisting of 19 temperature- and precipitation-related variables of the WORLDCLIM dataset [25], (ii) the BIOCLIM35 (P2) including the BIOCLIM19 variables and 16 additional radiation-, water-balance and soil-moisture-related variables [26] and (iii) the BIOCLIM35+T4 (P3) in which a set of four topographic variables derived from a digital elevation model (DEM), downloaded from the WORLDCLIM data portal [27], was added. The slope, aspect and … Appendix A. Detailed information on the development of the P1 and P2 datasets is given in Hijmans et al. [27].

>> Updated accordingly

(Note: Are the two [27] correct?)

>> Thank you. The first one refers to the BIOCLIM download page and the next one to the WORLDCLIM. Both were pointing to the BIOCLIM page reference, we have now corrected this. 

Species data. Remove (TD, biotic)

You could combine the ‘Presence data’ and ‘Pseudo-absence data’ as follows:

The worldwide distribution of five insect species, (i) Aedes albopictus (Skuse, 1894), (ii) Diabrotica virgifera virgifera (LeConte, 1868), (iii) … was recorded using presence-only data acquired from three sources, (a) the Global Biodiversity Information Facility (GBIF) database, (b) previous literature and (c) personal communication with relevant experts (Fig. 2).

>> Most of the text under the “Presence data” section is updated as per the direction above.

Pseudo-absences were generated from the presence-only data using a 3-step … [19]. (Remove ‘This method … presence points’ -unnecessary information-). Then describe the pseudo-absence data generation method. I got somewhat confused in lines 145-156, but see this article, it is far more comprehensive on how the pseudo-absence data are generated

https://ac.els-cdn.com/S030438001500215X/1-s2.0-S030438001500215X-main.pdf?_tid=e892b825-d0c7-48df-8ae6-8a81a84581f5&acdnat=1547468925_9b8d610c1322d75cfe050f750d9f3a94

>> The Senay, et al. [1]paper uses a slightly different method to determine the appropriate distance at which the background data is truncated. Additionally, that was where the method was first described. However, we have also added a citation for Iturbide, et al. [2]so readers can have access to both. We agree there are times when the Iturbide, et al. [2]method is easier to generate pseudo-absences using the 3-step method especially when the study area is small and it becomes difficult to observe variation in correlation between variables over small distances.

Remove all unnecessary info, e.g. since you provide the species name, the class and family names are not necessary, we can search for them if we don’t know them.

>> Done

(Note: You say ‘invasive’. Invasive to where, since your study area is the whole planet?)

>> We have now added a reference to Note S4 that describes the native and invaded range of each of the five species.

Lines 125-129. These lines are also not informative and they complicate things instead of easing the understanding. I would remove them. The reader can see Fig. 2 and see the distribution of each species.

>> We have kept the explanation about the ROA variation of the different occurrence datasets as it is an important factor to explore in terms of whether variation in ROA affects model predictive accuracy. We agree that this is already visible in Figure 2. We wanted readers to know that the variation was intentional and not coincidental. We have added a statement, “The variation in ROA among these species is important to determine if distribution range of a species affects the predictive accuracy of SDMs.” to clarify why we brought up the subject (ROA of the species) here.

Lines 131-135. Why do I, as a reader, need to know about the species established in New Zealand? At least it is not necessary for this specific article whose focus is on the model comparison and not on the species distribution.

>> Removed

Lines 137-140. Topographic map showing the global distribution of five insect species selected for the study: (A) Aedes albopinctus, (B) … I don’t understand the ‘relative global extent’. Relative to what? To the global map? To the other species coverage? I don’t think it is necessary here anyway.

>> We have changed the description of panel (G) to “The geographic extent of each species”. We believe this makes more sense now that the importance of variation in ROA for the factorial study has been explained. However, we would remove it in the suggested second round review if the reviewer feels that it still is not well justified.

Dimension reduction. Remove (DR)

>> Done

Three dimension reduction methods were used:

1. Variable selection using the Random Forest algorithm - RF (DR1): The RF algorithm can handle large numbers of variables and it is widely used in species distribution modelling … The Akaike’s Information Criterion (AIC - reference) was used to …

2. Principal Components Analysis - PCA (DR2): The PCA is a …

3. Hierarchical non-linear PCA - NLPCA (DR3): This method is a neural network model developed by … Remove ‘While there are many … [35]’ and start with ‘This method has been reported to be …’.

>> Edited accordingly

Lines 203-206. This is not my expertise so, just make sure you have run the method appropriately.

>> Thank you, the network initializing parameters given in those lines are correct.

Model selection

An exhaustive comparison of all the SDMs available would be impractical. In this study, we selected the four following models-algorithms to develop predictions on the global distribution of each species based on the various P and DR combinations:

1. Quantitative Discriminant Analysis - QDA (MT1):

2. Logistic Regression - LOGR (MT2):

And describe the models similarly to the aforementioned pattern (simple and without unnecessary information).

>> Edited accordingly. We kept the section title as “Model type” for the sake of consistency.

Model choice evaluation. Table 3 is not necessary, you can just add the metrics and the references in the text.

>> Table 3 Removed

A multiple-factor multivariate … the effects of TD, P, DR and MT on five metrics that are commonly used to measure model performance, Kappa (reference), AUC (reference), Sensitivity-Specificity scores (reference) and the cross-validation error (RMSE – reference). MANOVA was used to make ….

‘Model ranking’ can be merged with the previous subsection and start after … multcomp [51]. The model with the maximum observed metric based on the MANOVA …

>> Edited accordingly

RESULTS

Line 263-273. All this information (Mahalanobis’ distance, Chi-squared, Box’s M) and the justification of MANOVA should be mentioned in the Materials and Methods section, in the subsection talking about the MANOVA.

>> The section is now moved to methods

Line 273. Practically, your results start from here.

>> We agree, the result for this section now starts from this line

Line 276. MANOVA-results table showing the effects of the various modelling components on model performance

>> Done

Line 276. In the revised version, take care about the squared values (χ2, η2) because they appear as χ2, η2 … Also, you can just write ‘η2 (%)’ instead of using a footnote.

>> Edited accordingly

Line 276. I am not an expert in MANOVA but shouldn’t there be a probability value showing the significance of each component? Like this one for example …

https://stats.stackexchange.com/questions/215155/how-to-read-the-output-from-manova

>> We absolutely agree. This column was left out by mistake in our MS. We have added the Probability column in the results now.

Lines 278-286. Again, based on this one https://ncss-wpengine.netdna-ssl.com/wp-content/themes/ncss/pdf/Procedures/NCSS/Canonical_Correlation.pdf there should be a probability value mentioned.

>> This information is now Included for the first canonical variable reported in the results section. Additionally, we now include all the results of the correlation analyses in a new supporting information table (Table S1) along with our updated submission.

Line 280. 81.5% (0.9032 x 100)?  

>> Thank you, for bringing this to our attention! This was a mistake. We wanted to report that the canonical coefficient of determination (R2) for the first dimension is calculated by obtaining the square of the canonical correlation coefficient for the first canonical dimension. It is now corrected to 0.90342. As we are reporting correlation (R2) the percentage was included by mistake. It is now removed.

Figure 4. Are you sure the red line and circle helps? What is the green line showing? If not removed, you should mention what these lines and circles depict.

>> Thank you for bringing this to our attention. We have added an explanation about all the elements given in the graph. We have also colored the data ellipse blue instead of red, to avoid confusion with the linear regression line (red), the locally weighted scatter plot smoothing fit is shown in green and the centroid for 68% of the data point with in the 1st SD is shown in solid black dot.

Line 298. You didn’t say in the Materials and Methods that you applied hierarchical partitioning. Please write down all the methods and analyses you used in the relevant section and report here only your results. I would write (in the Materials and Methods after line 246), ‘hierarchical partitioning was carried out using the hier.part package […] to quantify … ‘.

>> This is now moved to the methods section on L262-265 of the updated MS.

Also, since you defined abbreviations for all your inputs (e.g. species data (TD), predictor dataset (P) etc.) use only the abbreviations here. You may use the ‘species data’ etc in the discussion.

>> Done

Line 308. Again, you have already defined DR, don’t define it again and again, just use DR. Use only the abbreviations throughout.

>> Done

Lines 325-328. These are for the discussion section.

>> Removed and moved to L472-475 of the updated MS

Line 331. Have you described the method of Dormann et al. in the Materials and Methods?

>> Thank you! No it was not described in Methods, we have now moved this description to L266 –L270 in the Methods section of the updated MS.

Lines 330-334. These lines are not results, they should be in the Materials and Methods section.

>> Same as above, the part that describes this method for the first time is now moved to the methods section L266 –L270 of the updated MS.

Lines 339-353. Use the abbreviations for the model types, you have already defined all these models before.

>> Done

Line 343. Which score? The Kappa or the CV?

>> Kappa, we have now clarified this.

Why not depicting all these Kappas and CVs for the various DR methods using a bar chart? Or a table at least?

>> Mean Kappa scores for all TD-DR-MT combinations are given in Table S4.1, this is now indicated within the section (L405 of the updated MS).

Line 370. Best and worst component combinations for the five species modelled in this study. For the abbreviations used, please refer to Fig. 1.

>> Done

Line 380. Fig. 5 would be more informative if the various dots (representing models) had different shapes. I assume it will be visually more complicated, but you could apply different shapes for the four MT at least, if not for each P-DR-MT combination.

>> Graph (Figure 5) updated as per the direction above.

Reference:

1.             Senay, S.D.; Worner, S.P.; Ikeda, T. Novel Three-Step Pseudo-Absence Selection Technique for Improved Species Distribution Modelling. PLoS ONE 2013, 8, e71218, doi:10.1371/journal.pone.0071218.

2.             Iturbide, M.; Bedia, J.; Herrera, S.; del Hierro, O.; Pinto, M.; Gutiérrez, J.M. A framework for species distribution modelling with improved pseudo-absence generation. Ecological Modelling 2015, 312, 166-174, doi:https://doi.org/10.1016/j.ecolmodel.2015.05.018.

Round 2

Reviewer 2 Report

Dear authors,

this version is much upgraded!

Please see my comments (attachment) on your second version, I think that after a detailed addressing of these comments your article will be ready. So, I currently can't select 'minor revision' but a detailed revision will obviously result in acceptance from my side.

Kind regards,

Author Response

Response to comments of Reviewer 2 Round 2

Comments and Suggestions for Authors

Dear authors,

this version is much upgraded!

Please see my comments (attachment) on your second version, I think that after a detailed addressing of these comments your article will be ready. So, I currently can't select 'minor revision' but a detailed revision will obviously result in acceptance from my side.

Kind regards,

Dear authors and editors,

this version of the article is much upgraded and the manuscript is now more easily digested. The authors addressed nearly all my suggestions, thank you for this. I also read in the authors’ response that they will go through the English usage after the second revision round and yes, this needs to be done before publication. Also, there are still some details that need to be addressed, for example, abbreviations should be defined once and then used as abbreviations but this has not been currently applied throughout the text.

Scientifically, I think the Results and the Discussion section are generally ok. I provide some suggestions once again and apart from the English usage, and please (the most important!) check my questions regarding the conclusions and some parts in the discussion, which I think that need some re-thinking. It’s up to the authors to conclude and highlight the most important findings of their study, but before reading the conclusion I had some extra findings in my mind that were not included by the authors in the conclusions. So, re-check your discussion and conclusions accordingly.

I think that with the following suggestions and a thorough English re-checking, the article will be ok.

>> Dear reviewer, we are writing this with the highest recognition of the effort you put into our manuscript. We are very happy with how the article turned out following your thorough guidance. We also thank you for highlighting some results we did not discuss. We have also fixed all the duplicated abbreviation definitions we failed to remove in the first review. We hope you find the revision in this round acceptable. Once again, we are very grateful for your stellar review work. Please find our point-by-point response to your comments below.

ABSTRACT

The Abstract is now easily read. Consider though including some findings from my suggestions for the conclusion section.

>> We updated the abstract with most of the items suggested for inclusion in the abstract and the conclusion sections.

Line 13. Based on this https://forum.wordreference.com/threads/criticism-against-towards.634212/ you should write ‘There is also much criticism of prediction discrepancies among …’.

>> Done

Line 14. … and of the lack of …

>> Done

Lines 15-16. It’s not ‘species-training-datasets’ but ‘species -training- datasets’, that means, species datasets used for model training.

>> Done

Line 16. … dimension-reduction methods …

>> Done

Line 17. … of the SDM predictions … with the aim to …

>> Done

Line 18. Considering that in line 16 you said ‘modelling algorithms’ and here you say ‘model type’, I suggest to use the same terminology, at least in the Abstract. I would replace ‘modelling algorithms’ above, with ‘model types’.

>> Done

Line 18. … species-distribution predictions …

>> Done

Line 21. This sentences should be ‘We also found that the widely used confusion-matrix-based model-performance indices …’ (see here https://getitwriteonline.com/articles/hyphenated-adjectives/ and here https://www.grammarbook.com/punctuation/hyphens.asp for correctly hyphenating, they are very useful!).

>> Done

Line 22. … Sensitivity, Kappa and others … Also, I think ‘discriminate’ requires something like ‘between’, ‘against’ etc., so, maybe replace with ‘select’.

>>Changed to select

Line 23. … model-result discrepancies …

>>Done

Line 24. … necessarily suggest … Also, I think it’s either ‘lack of robustness in correlative modelling’ or ‘lack of robustness of correlative models’.

>> Done

Line 25. ‘selection’ instead of ‘choice’?

>> Done

Lines 25-26. ‘Our results … signal … research on model-performance evaluation …

>> Done

Line 27. … model-evaluation methods …

>>Done

Line 29. … species-distribution modelling, where possible, is likely … mitigate errors … from the selection of inappropriate components.

>> Done              

Line 30. It also provides end users with … (no hyphenation is necessary here but it is necessary in … model-prediction uncertainty …).

>> Done

Line 31. … from the selection of inappropriate components …

>> Done

I would write the concluding sentence (to avoid ‘modeller’s choice’ which I don’t think it can be written this way) as follows:

‘Therefore, undertaking multi-scenario species-distribution modelling, where possible, is likely to mitigate errors arising from inappropriate modelling-components selection and provides end users with better information on the resulting model-prediction uncertainty’.

>>Done, this reads better.

INTRODUCTION

The Introduction has also been upgraded and is now easily read. Some minor comments, again, to be considered.

Line 37. Maybe ‘species presence along with environmental data’ (?)

>> Done

Line 39. … is a major issue …

>> Done

Line 41. … in the predictive accuracy …

>> Done

Line 42. Is it ‘model algorithms’ or ‘modelling algorithms’?

>> The latter is better, now edited.

Line 44. … range, representing … Also, I don’t understand ‘limited environmental range’… do you mean ‘limited environmental data’ or ‘limited geographical range’? Finally, what is an ‘uncomplicated interaction’?

>> We wanted to indicate the size of the environmental niche, in other words, the locations that fulfill the physiological requirements of the species as given by the optimum values of the appropriate environmental variables. For example, an optimum combination of a given temperature and precipitation range that is required for the survival of a given species. We have now changed it to “occupying limited environmental niche”.

>> Uncomplicated now changed to linear

Line 44. Remove ‘While’?

>>Done

Line 55. … has been reported … (as it refers to the use).

>> Done

Line 58. …information, such as elevation, that …

>> Done

Lines 61-63. I’m not sure I understand what you want to say here… what is a multi-source variable and a variable from the same source?

>> A list of variables compiled from different sources. For example, data sourced from different remote sensors would require standardizing formats, checking if bands are from the same wavelengths etc. We have now switched “multi-sourced” to “data compiled from different sources” for clarity.

Line 63. Insert a comma after ‘increase’.

>> Done

Line 64. … the scale … differ; this increases modelling …

>> Done

Line 65. … complication of including increased number …

>> Done

Line 66. … is the increased … among large sets …

>> Done

Line 68. Remove ‘which made using simple models possible’.

>> Done

Line 68. Just a suggestion… ‘However, in ecology, linear relationships are not always observed between environmental variables [24]. This becomes specifically apparent large sets of predictors from multiple scales and data sources are used in the modelling process’.

>> Done

Line 71. … is used … (as it refers to ‘a large number’)

>> It is now referring to ‘large sets’ as per the suggested phrase above, so we kept this as it is.

Line 71. … Thus, the type and number of variables used, further …

>> Done

Lines 72-74. … between complex and simple models due to the inclusion or not of the interactions among the environmental variables [2,17].

>> We note your suggestion but changed “inclusion” into “accurate representation” as it is not the lack of accounting for interaction that creates the difference but how the interaction is represented. For example, representing a non-linear interaction by a linear relationship.

Line 76. … predictor datasets …

>> Done

Line 77. … dimension-reduction methods … the same applies to line 81 and line 85.

>> Done

Line 82. … can be successfully modelled after the …

>> Done

Line 84. … species-occurrence data …

>> Done

Line 86. … could affect the performance …

>> Done

Lines 87-89. I think these lines fit better to the end of the first paragraph, after line 40. It would be then technically more correct that your final paragraph starts with ‘The purpose of this study …’, as this is what the reader seeks at this last paragraph.

>>We agree, updated accordingly.

Lines 90-91. … such as predictor datasets, dimension-reduction methods, … on the SDM model performance …

>> Done      

Line 93. … appropriate predictors, dimension-reduction methods and model types that …

>> Done

Line 95. … we discuss on the advantage … to provide information on model prediction uncertainty.

>> Done

MATERIALS AND METHODS

Line 99. Remove ‘mainly’?

>> Done

Line 100. … species-presence datasets, however, cover …

>> Done

Line 113. … DEM-derived …

>> Done

Line 132. … dimension-reduction … with the Random Forest …

>> Done

Line 134. … low-bias …

>> Done

Line 135. … but low-correlation trees …

>> Done

Line 136. … Criterion …

>> Done

Line 140. … dimension-reduction …

>> Done

Line 142. … the non-linear extension …

>> We needed “true” here as there are other non-linear extensions of PCA but they do not accurately emulate the way the orthogonal dimensions are derived and ordered in PCA.

Line 148. … was minimum …

>> Done

Line 154. … (a) the Global …

>> Done

Line 156. ROA should be defined here instead of line 158.

>> Done

Line 167. … using a 3-step …

>> Done

Line 209. … we selected four model types … (or modelling algorithms if you prefer).

>> Done, chose model types for consistency

Line 255. … were analyzed using a single factor analysis of variance (ANOVA) with Tukey’s honestly significant difference (HSD) post-hoc test.

>> Updated accordingly.

Note: In lines 246-247 you define abbreviations about dimension reduction etc. but in lines 254-255 you use again the normal words instead of the abbreviations. Also, the abbreviations are also defined in line 242 and again in 246-247. The normal way to do this would be to define the abbreviations only ONCE (for example in the Research Design and Model Conceptualization section, which should be transferred after line 103 and then use only the abbreviations throughout the text). Otherwise there is no need to define an abbreviation and not use it.

>> This is corrected now

Lines 276-290. These lines (and the relevant figure, which should become Figure 1 instead of 3) are very well written, showing an overview of the research method, but they should be placed after line 103.

>> We had this section at the suggested location in the previous version of our MS, however the other reviewer suggested that for people who want to follow the design in detail, it would be much easier to have the summary of the research design after each of the components are described. That is why we moved it as a summary of the methods section. We hope this also works. We will move the lines back if you believe that being at the beginning has more merit than its current location at the end of the methods section.

>> However, we have moved the section in front of the “model choice evaluation” section to avoid defining the modelling components twice (once in the research design and again in the model choice evaluation). We have used the abbreviations throughout the “model choice evaluation” section now.

RESULTS

Line 293. … MANOVA results …

>>Done

Line 296. … on model performance.

>>  Here we gave a list  of model performance scores analyzed  using MANOVA,  rather than summing them up as “model performance”. We did tis so readers can clearly follow what elements were involved in the MANOVA.

Table 3. Delete the abbreviations (MT, DR, P, SP) since you don’t use them in the Table. Also, the asterisks and the ‘ns’ are not necessary, you can just add the asterisks as a superscript to the p<0.001 column.

>> Done

Line 302. There is an extra 2 after 0.9034.

>> That is to show that we squared the R-value (0.9034) to obtain the R^2 which is 0.816 (0.9034^2).  We have now removed the notation showing the actual R being squared and just gave the R^2 as it was redundant.

Line 334. … PCA-based …

>>Done

Line 342. Remove ‘Species’.

>> Done

Line 349. Mainly instead of most?

>> Done

Line 368. Here, again you define SVM, CART, LOG etc. They have been already defined in the Materials and Methods, so just use the abbreviations here. Correct accordingly throughout the text please.

>>Done

Line 421. Table 4.

>>Done

DISCUSSION

Lines 457-458. … model type, dimension-reduction method, species dataset and predictors dataset on …

>>Done

Lines 459. … sources of uncertainty …

>>Done

Line 464. … requiring a case-by-case …

>>Done

Line 468. Again, do you mean limited geographical distribution?

>> Changed this to “environmental niche”

Lines 475-485. And here again you use the already defined abbreviations with the relevant names. Use only the abbreviations. BIOCLIM35 (P2) [31] for example is totally wrong. You have already defined BIOCLIM35 as P2 and now you use both the name and the abbreviation.

>>Done

Also, I don’t understand some parts here. Are you referring to Table S2 and Table 1? All first 5 or 6 variables were both in the P1 and P2 datasets. Which were the individual variables that were only in P1 and were included in the models? You can write some of these variables in Lines 476-477. Again, which were the second most included set of variables? In the Results you only refer to the first 5 ranked variables which I see that they are included in both P1 and P2. Thus, I understand the suggestion of ‘not P3’ but why to propose P2 instead of P2?

>>  We think we have now described it better. The text in L461-473 in the updated MS now shows that most of the top half of the ranked list of variables are those that are found in P1 and are common to all P1,P2 & P3, therefore showing P1 is an important dataset. However, some of the variables ranked in the top half of the list uniquely belong to P2 & P3, but none that were unique to P3 were included in the top 50% of the ranked order. Therefore, we recommended using the P2 dataset unless there is expert knowledge that justifies the use of temperature and precipitation based variables alone.

Lines 537-539. The fact that a model had lower (although not significantly lower) Kappa means that it predicts slightly worse than the other with the higher Kappa. Or not? OK about the ‘similar’ but can you say that the model with a lower Kappa may give better predictions?

>> We specified a possible reason for the latter to happen by writing it as follows “….similar or sometimes better (in case of models that over-fit and or extrapolate) predictions than the absolute highest Kappa score model”.

>> We think this might be an important issue to mention as in most cases the training data does not cover the whole geographic extent of the areas for which species distribution is predicted (especially in large scale studies like this). Which means we have very little evidence to confirm if, for example, one missed correctly predicting just one test occurrence point which lead to a slightly lower score, would translate into a significantly inaccurate prediction in the areas that are not covered by the training data.

Similarly, if the second best model provided a better representation of the relationships between the chosen predictors and projected a better species distribution prediction for areas not covered by the training data, even if it missed accurately predicting more test data points than the best model, it is a possibility that it might have a better prediction, as long as the scores between the best and the second best models are not statistically significant.  The slight under-performance might be a result of being penalized by fitting the known dataset less accurately at the expense of fitting the extrapolated environmental space better. This is largely discussed as a case of SDM extrapolation [1 and references therin,2]

Our result has shown that the Kappa score consistently discriminated between the best and worst models. Our message here is that currently used evaluation methods are not sensitive enough to discriminate between high performing models whose scores are not significantly different.

CONCLUSIONS

Just start with your results and conclude your research focusing on highlighting the most important findings. It would be correct not to include references here.

>> We removed the sentence “Because of that it has been continually called for in the literature” from L568 which had the references in this section.

And shorten the conclusions to just one paragraph.

>> We shortened the conclusion section  as per the suggestion

So, I think that your most important findings are the following (please consider mentioning some of these conclusions in the Abstract also):

>> Abstract updated to include most of the items listed below.

1. A model’s predictive performance is highly influenced by data pre-processing; pseudo-absence dataset, dimension reduction, species dataset, predictors’ dataset and model type of course.

2. Unless all other factors (TD, PD, P) are kept constant, you can’t effectively compare the performance of different MTs

3. With constant TD, PD, P, machine learning models seem to outperform the other model types

4. h-NLPCA-based models also showed highest performance

5. Low-occurrence species can be effectively modelled by less complex models (I think I read this in the discussion).

‘This study showed that the predictive performance of different model types depends much on data pre-processing, that is, on pseudo-absence dataset development, dimension-reduction method, species- and predictor-dataset selection. Thus, performance comparisons among different model types cannot be applied unless all data pre-processing factors are kept constant. The model-type comparisons applied for the same species and predictor datasets and under the same dimension-reduction method showed that machine learning models (mainly SVM) outperform other model types. The h-NLPCA was also a highly ranked dimension-reduction method’.

>> This is a very good wrap up of our findings. We liked it and included it in our conclusion. We also felt that the finding about the evaluation methods is important so we added that following your summary. Thank you.

And then you can continue concluding about the use of multi-scenario modelling frameworks. Use less words and state clearly your suggestions (remember you are now concluding, that is, you are nearly finishing your article). But I leave it up to you to conclude the article in the way that you think is most suitable. But follow the concept I propose, it is clearer and will be easier to read from someone else.

>> We have shortened the portion of the conclusion that comes after the section you summarized. In other words, the conclusion about the benefit of using multi-scenario modelling and using a second evaluation method. We have also edited the last portion that concludes about how the information on model uncertainty obtained from the multi-scenario modelling framework maybe used in real-world applications supporting sampling methods used for invasive insect species detection.  

Note for example that lines 544-549 are not conclusions but discussion.

>> Removed

SUPPLEMENTARY MATERIAL

1. It is very helpful that you provide all these results.

I think that all these abbreviations in the caption of Table S1.1 could be defined in the first page of the supplementary material (BUT if you think it is too much it’s ok, I’m just trying to make the article as easy-to-be-followed as it gets).

For example:

SPaa: Species dataset, Aedes albopictus

SPag: Species dataset, Anoplopis gracilipes

….

DRdr2: Dimension reduction, Principal Components Analysis

MTlog: Model type, logistic regression

>> >> We agree, we have now provided the definition suggested. We have also changed all references to SP into TD to keep it consistent with the main manuscript.

2. Figure captions should be below the figure, table captions above the table.

>> Done both in Figure S1.4, Figure S3, Figure S4.2, all figures in S6, S7 and S8

3. Please harmonize all abbreviations like (Fig. 6.c). In the Results you say Fig. 6C, here you say Fig.6.c. It’s not that the reader will not understand it, it’s a matter of being accurate.

>>Done

4. Instead of ‘Literature cited’, you could just write ‘References’.

>> Done both in Table S2 and Note S5

5. Table S2. Why not place the variables in the table based on their ranking? Starting from Annual precipitation, then Precipitation of the driest quarter etc.

>> Done

6. Table S4.1. In the caption you write DR but in the table you write dr. Also, you say species data (SP) but there is no SP in the table but the species names abbreviated. Check for such mistakes throughout the text before publication please.

>> Keys similar to Table S1.1 and S1.2 are now provided for Table S4.1.

APPENDICES

Appendices A and B are not necessary for your article. Everyone has an idea of how slopes are calculated and even if he/she hasn’t he/she can search for such essential information. Also, the possible audience of your article are also aware of the importance of reducing dimensions.

>> Done

You could only keep Appendix C (which would now be A) since the h-NLPCA is a relatively new method. Note that your article is already long and from my experience, if an article exceeds 8 pages (in the journal’s version) the reader will either go through it very quickly or just stop reading after the 10th page.

>>Done

Reference

1.            Elith, J.; Phillips, S.J.; Hastie, T.; Dudík, M.; Chee, Y.E.; Yates, C.J. A statistical explanation of MaxEnt for ecologists. Diversity and Distributions 2011, 17, 43-57, doi:10.1111/j.1472-4642.2010.00725.x.

2.            Raes, N.; Aguirre‐Gutiérrez, J. Modeling Framework to Estimate and Project Species Distributions Space and Time. Mountains, Climate and Biodiversity 2018, 309.

Link to Reference 2: (https://www.researchgate.net/profile/Jesus_Aguirre-Gutierrez/publication/323801521_A_Modeling_Framework_to_Estimate_and_Project_Species_Distributions_in_Space_and_Time/links/5b3100a8aca2720785e4b739/A-Modeling-Framework-to-Estimate-and-Project-Species-Distributions-in-Space-and-Time.pdf)

Round 3

Reviewer 2 Report

Dear Authors,

this version seems scientifically and technically ok.

I attach your text with some minor corrections.

Best regards

Author Response

Response to comments of Reviewer 2 Round 3

Comments and Suggestions for Authors

Dear Authors,

this version seems scientifically and technically ok.

I attach your text with some minor corrections.

Best regards

>> We are very grateful for all your input in improving our manuscript throughout the peer-review process. We have now made the minor updates you suggested in your third round review. We have indicated the changes we have made to the manuscript through a track-change. We have also attached the PDF you used to leave us suggestions, we replied to each suggestion within the comments you made on the PDF. Thank you.
